# Sex differences in the longitudinal relationship of low-grade inflammation and echocardiographic measures in the Hoorn and FLEMENGHO Study

**Sharon Remmelzwaal**[1]*, **Joline W. J. Beulens**[1,2], **Petra J. M. Elders**[3], **Coen D. A. Stehouwer**[4], **M. Louis Handoko**[6], **Yolande Appelman**[7], **Vanessa van Empel**[8], **Stephane R. B. Heymans**[9,10,11], **Lutgarde Thijs**[5], **Jan A. Staessen**[5,6], **A. Johanne van Ballegooijen**[1,12]

1 Department of Epidemiology & Data Science, Amsterdam UMC, Amsterdam Cardiovascular Sciences, Vrije Universiteit Amsterdam, Amsterdam, The Netherlands, 2 Julius Center for Health Sciences and Primary Care, University Medical Center Utrecht, Utrecht, The Netherlands, 3 Department of General Practice and Elderly Care Medicine, Amsterdam Public Health Research Institute, Amsterdam University Medical Center, Location VUmc, Amsterdam, The Netherlands, 4 Department of Internal Medicine, Cardiovascular Research Institute Maastricht (CARIM), Maastricht University Medical Center+, Maastricht, The Netherlands, 5 Research Unit Hypertension and Cardiovascular Epidemiology, KU Leuven Department of Cardiovascular Sciences, University of Leuven, Leuven, Belgium, 6 NPA Alliance for the Promotion of Preventive Medicine (APPREMED), Mechelen, Belgium, 7 Department of Cardiology, Amsterdam UMC, Vrije Universiteit Amsterdam, Amsterdam, The Netherlands, 8 Department of Cardiology, Maastricht University Medical Center, Maastricht, The Netherlands, 9 Department of Cardiology, CARIM School for Cardiovascular Diseases Faculty of Health, Medicine and Life Sciences, Maastricht University, Maastricht, The Netherlands, 10 Centre for Molecular and Vascular Biology, KU Leuven, Leuven, Belgium, 11 Holland Heart House, ICIN–Netherlands Heart Institute, Utrecht, The Netherlands, 12 Department of Nephrology, Amsterdam UMC, Amsterdam Cardiovascular Sciences, Vrije Universiteit Amsterdam, Amsterdam, The Netherlands

* S.Remmelzwaal@amsterdamumc.nl

**Data Availability Statement:** Data cannot be shared publicly because of restrictions in informed consent of both cohorts. For the Hoorn Study, data

## Abstract

### Background

This study aimed to determine the within-person and between-persons associations of low-grade inflammation (LGI) and endothelial dysfunction (ED) with echocardiographic measures related to diastolic dysfunction (DD) in two general populations and whether these associations differed by sex.

### Methods

Biomarkers and echocardiographic measures were measured at both baseline and follow-up in the Hoorn Study (n = 383) and FLEMENGHO (n = 491). Individual biomarker levels were combined into either a Z-score of LGI (CRP, SAA, IL-6, IL-8, TNF-α and sICAM-1) or ED (sICAM-1, sVCAM-1, sE-selectin and sTM). Mixed models were used to determine within-person and between-persons associations of biomarker Z-scores with left ventricular ejection fraction (LVEF), left ventricular mass index (LVMI) and left atrial volume index (LAVI). These associations were adjusted for a-priori selected confounders.

are available from the Ethics Board of the Hoorn Studies (contact via hoornstudy@vumc.nl) for researchers who meet the criteria for access to confidential data. For FLEMENGHO, anonymized data can be made available to investigators for research based on a motivated request to be addressed to the Ethics board of the Hypertensive and Cardiovascular Epidemiology department via infohypergenes@kuleuven.be. Data of both cohorts are stored on independent secured network locations within both hospitals. These network locations have daily back-ups.

**Funding:** This work was supported by the Netherlands Cardiovascular Research Initiative: an initiative with support of the Dutch Heart Foundation (CVON2016-Early HFPEF and CVON2017- She-Predicts). SR and AJvB are supported by a ZonMw grant (849500008). JWJB is supported by a ZonMw VIDI grant (91718304). SRBH received support of the ERA-Net-CVD project MacroERA (01KL1706) and IMI-2CARDIATEAM (821508). Furthermore, we acknowledge the support of FWO (Belgium; G091018N & G0B5930N) to SRBH. The NPA Alliance for the Promotion of Preventive Medicine (APPREMED), Mechelen, Belgium received a non-binding grant from OMRON Healthcare Co. Ltd., Kyoto, Japan.

**Competing interests:** The authors have declared that no competing interests exist.

## Results

Overall Z-scores for LGI or ED were not associated with echocardiographic measures. Effect modification by sex was apparent for ED with LVEF in both cohorts (P-for interaction = 0.08 and 0.06), but stratified results were not consistent. Effect modification by sex was apparent for TNF-α in the Hoorn Study and E-selectin in FLEMENGHO with LVEF (P-for interaction ≤ 0.05). In the Hoorn Study, women whose TNF-α levels increased with 1-SD over time had a decrease in LVEF of 2.2 (-4.5;0.01) %. In FLEMENGHO, men whose E-selectin levels increased with 1-SD over time had a decrease in LVEF of 1.6 (-2.7;-0.5) %.

## Conclusion

Our study did not show consistent associations of LGI and ED with echocardiographic measures. Some evidence of effect modification by sex was present for ED and specific biomarkers.

## Introduction

As the prevalence of heart failure (HF) is increasing, this common disease is an emerging public health problem [1, 2]. Over half of the HF patients have heart failure with preserved ejection fraction (HFpEF), a syndrome characterized by a normal ejection fraction and elevated left ventricular filling pressures [3]. The underlying pathophysiological mechanisms of HFpEF are unclear, and patients with HFpEF are not featured by one phenotype and are predominantly female [4]. The current treatment options for HFpEF are limited in contrast to patients with heart failure with reduced ejection fraction (HFrEF) with a more distinct clinical presentation [5–7].

Recently, a new paradigm has been proposed where systemic inflammation and endothelial dysfunction is induced by comorbidities, causing structural changes with consequent HFpEF development [8]. Obesity, hypertension and type 2 diabetes are the most common comorbidities, that lead to increased levels of inflammatory biomarkers such as interleukin-6 (IL-6) and tumor necrosis factor-α (TNF-α). Higher levels of these inflammatory biomarkers would trigger a myriad of reactions, eventually leading to stiffer cardiomyocytes and left ventricular diastolic dysfunction (LVDD).

LVDD is a precursor of HFpEF, and both phenotypes are observed more frequently in women than in men [9, 10]. In both phenotypes, inflammatory biomarkers are elevated compared with healthy adults [11–16]. Inflammation could play a role in the sex differences of LVDD and HFpEF prevalence. In population-based cohorts, Framingham Heart Study and the National Health and Nutrition Examination Survey (NHANES), and Utrecht Coronary Biobank (UCORBIO) that included patients who underwent coronary angiography, women had higher C-reactive protein (CRP) levels compared to men [17–19]. In addition, higher CRP levels were associated with higher left ventricular mass in German women of the population-based CARLA study, but not men [20]. IL-6 levels were not associated to any echocardiographic measures in both women and men [20].

To date the association between specific inflammatory biomarkers and echocardiographic markers related to LVDD has been studied only in cross-sectional settings and specific patient populations. It is unclear if a spectrum of inflammatory or endothelial dysfunction biomarkers together is associated with the development of LVDD and HFpEF and could explain the

differences between women and men and their predisposition to develop LVDD and HFpEF. Therefore, this study aimed to determine the within-person and between-persons associations of low-grade inflammation and endothelial dysfunction, with echocardiographic markers related to DD in two general populations and whether these associations differed by sex.

## Materials and methods

The current study included participants with baseline and follow-up visits of two cohorts: the Hoorn study and the Flemish Study on Environment, Genes and Health Outcomes (FLEMEN-GHO) [21, 22].

### Study population

**The Hoorn Study.** The Hoorn Study is a prospective cohort, that started in 1989, and was initiated to study the prevalence and determinants of type 2 diabetes in the general population of Hoorn, the Netherlands. For the Hoorn study, we included 831 participants who underwent echocardiographic measurements in 1999–2001, considered as baseline. This subgroup was oversampled for participants with impaired glucose metabolism (IGM) and type 2 diabetes (T2D) to enable investigation of effect modification by glucose metabolism status [21]. We excluded 102 participants with missing information on low-grade inflammatory or endothelial dysfunction biomarkers (N = 64), missing left ventricular ejection fraction (LVEF), left ventricular mass index (LVMI) or left atrial volume index (LAVI) (N = 35), or both (N = 3). During follow-up, 111 participants died, 49 had physical or mental health problems and 39 were untraceable or moved out of the area, leaving 522 participants, 407 (78.0%) of whom underwent repeated echocardiographic measurements in 2007–2009. We excluded 24 participants with missing on low-grade inflammatory or endothelial dysfunction biomarkers at follow-up (N = 4) or with missing LVEF, LVMI or LAVI available at follow-up (N = 20), resulting in an analytic sample of 383 participants. We did not have complete data on all three echocardiographic outcomes, which resulted in different analytic samples for each outcome measure: 289, 304 and 318 participants for LVEF, LVMI and LAVI, respectively.

**FLEMENGHO.** FLEMENGHO is a family-based population study. From August 1985 until December 2015, we recruited a random sample of households living in a geographically defined area in Northern Belgium, as described elsewhere [23]. From 2005 to 2009, we invited 1,031 former participants for a re-examination at the field center, including echocardiography. Written informed consent was obtained for 828 participants (80.3%). We excluded 181 participants with missing information on low-grade inflammatory or endothelial dysfunction biomarkers (N = 171), with missing LVEF, LVMI or LAVI available (N = 7), or both (N = 3). During follow-up, 15 participants died, 10 had physical or mental health problems and 14 were untraceable or moved out of the area, leaving 608 participants, 520 (85.5%) of whom underwent follow-up echocardiography in 2009–2013. Participants with missing information on low-grade inflammatory or endothelial dysfunction biomarkers at follow-up (N = 26) or without either LVEF, LVMI or LAVI available at follow-up (N = 3) were excluded, resulting in an analytic sample of 491 participants. We did not have complete data on all three echocardiographic outcomes, which resulted in different analytic samples for each outcome measure: 315, 477 and 483 participants for LVEF, LVMI and LAVI, respectively.

All individuals gave written informed consent, and the ethical committees of the VU University Medical Centre, Amsterdam, The Netherlands and University of Leuven, Belgium approved the study. All data and samples from the study participants were fully anonymized before analyses.

## Low-grade inflammation and endothelial dysfunction measurements

For both cohorts, fasting, venous blood samples were drawn by trained nurses. In the Hoorn study, biomarkers of low-grade inflammation (C-reactive protein (CRP), serum amyloid A (SAA), interleukin 6 (IL-6), interleukin 8 (IL-8), tumor necrosis factor α (TNF-α) and soluble intercellular adhesion molecule 1(sICAM-1)), and of endothelial dysfunction (sICAM-1, soluble vascular cell adhesion molecule 1 (sVCAM-1), soluble endothelial selectin (sE-selectin) and soluble thrombomodulin (sTM) were measured, by a multi-array detection system based on electro-chemiluminescence technology (MesoScaleDiscovery, SECTOR Imager 2400, Gaithersburg, Maryland, USA). All serum samples were measured in duplicate. Intra- and inter-assay coefficients of variability were for CRP, 2.8 and 4.0%; for SAA, 2.7 and 11.6%; for IL-6, 5.6 and 13.0%; for IL-8, 5.6 and 12.2%; for TNF-α, 3.9 and 8.8%; for sICAM-1, 2.4 and 4.9%; for sVCAM-1, 2.8 and 5.6%, for sE-selectin, 2.6 and 6.7% and for sTM, 2.1 and 6.9%, respectively.

In FLEMENGHO, a single serum aliquot was used for all arrays. Therefore all biomarker panels were commenced within 1 hour of defrosting to minimize degradation. Levels of individual biomarkers were determined by Biochip Array Technology according to the manufacturer's instructions (Adhesion Molecule Array EV3519, Cerebral ArrayII EV3637 and Cytokine and Growth Factors Array (High Sensitivity) EV3623]. A broad pallet of biomarkers has been measured including: sVCAM-1, sICAM-1, sE-selectin, CRP, soluble tumour necrosis factor receptor I (TNFR1), vascular endothelial growth factor (VEGF), TNF-α, interleukin-1α (IL-1α), and epidermal growth factor (EGF).

## Echocardiographic measurements

In both the Hoorn study and FLEMENGHO, an experienced ultrasound technician or physician performed the echocardiographic examinations according to a standardized protocol consisting of two-dimensional, M-mode and pulsed-wave Doppler assessment [24–26]. An experienced cardiologist inspected the echocardiograms in order to ensure the quality. Baseline and follow-up echocardiograms were analyzed by the same technician or physician who was blinded to the participants' characteristics. LVEF (%) was used as an index of left ventricular systolic function and was calculated from the apical four chamber view in the Hoorn study [24, 25] and apical four- and two-chambers views in FLEMENGHO [27], using the modified Simpson's rule [28]. Left ventricular mass–measured in M-Mode and indexed to height to the power of 2.7 (LVMI, $g/m^{2.7}$)–was determined to assess cardiac structure. LAVI ($mL/m^2$) served as an index for LV diastolic function and in the Hoorn study was calculated by indexing single-plane maximum left atrial volume from the apical four chamber view by body surface area. In FLEMENGHO, left atrial dimensions were measured in 3 orthogonal planes (parasternal long, lateral, and supero-inferior axes) and LAVI was calculated using the prolate-elipsoid method [29] and was indexed to body surface area.

## Covariates

In both cohorts, study personnel collected demographic, metabolic and other risk factors with standardized methods during the baseline and follow-up visits. Smoking status was categorized in never smokers, former smokers and current smokers. Educational level was self-reported and was stratified into three categories: low (no or primary education), middle (secondary education) and high (tertiary education). History of cardiovascular disease (CVD) was based on self-report or medical records and was defined as any presence of angina, claudication, infarction, stroke, high ankle-brachial index, ECG abnormalities or arterial surgery. Weight and height were measured bare-foot and light-clothed, to calculate body mass index (BMI)

(kg/m2) (weight (kg) divided by height (m) squared). In the Hoorn Study, blood pressure (mmHg) was measured twice at the left upper arm in a sitting position using an oscillometric device and averaged (Collin Press-Mate, BP-8800). In FLEMENGHO, blood pressure was the average of five auscultatory readings obtained with a mercury sphygmomanometer after 5 minute rest in the seated position. Hypertension was defined as systolic blood pressure $\geq$140 mmHg or diastolic blood pressure $\geq$90 mmHg. All participants in the Hoorn Study, except those with previously diagnosed T2D, underwent a standard oral glucose tolerance test and were classified as either NGM (normal glucose metabolism), IGM (either impaired fasting glucose or impaired glucose metabolism), or T2D according to the 1999 WHO criteria) [30]. Estimated glomerular filtration rate (eGFR) (mL/min/1.73 m2) was calculated according to CKD-EPI 2009 [Chronic Kidney Disease Epidemiology Collaboration] formula [31]. Anti-inflammatory medication use (non-steroidal anti-inflammatory drugs (NSAIDs), platelet aggregation inhibitors (abciximab), lipid lowering agents (statins and niacins) and ACE inhibitors) was derived from questionnaires and were converted to ATC codes.

## Statistical analyses

**Z-scores.** Individual biomarker levels were combined into a Z-score of low-grade inflammation (CRP, SAA, IL-6, IL-8, TNF-$\alpha$ and sICAM-1) or endothelial dysfunction (sICAM-1, sVCAM-1, sE-selectin and sTM). Biomarkers with a right-skewed distribution (both cohorts: CRP, SAA, IL-6, IL-8, TNF-$\alpha$. FLEMENGHO: sE-selectin) were natural log-transformed. The baseline Z-scores were calculated for each biomarker as follows: (individual value–study population mean)/study population standard deviation (SD). The follow-up Z-scores were calculated for each biomarker as follows: (individual value–study population mean at baseline)/study population standard deviation at baseline (SD).

The Z-scores were averaged over all biomarkers to obtain an overall Z-score. Z-scores were calculated separately for baseline and follow-up, and for the Hoorn and FLEMENGHO studies. Higher scores indicate more low-grade inflammation or worse endothelial function, respectively. The Z-score represents the deviation from the mean per SD: a score of zero indicates no deviation from the mean, a score of one indicates that the biomarkers composing the Z-score are on average 1 SD larger than the mean.

**Within-between model.** All analyses were performed separately for Hoorn and FLEMENGHO. We fitted within-between models with a random intercept for both study populations by using the R-package 'panelr' because this model better allows to distinguish between between-person differences and change in biomarker Z-scores over time in individuals (within-persons differences) [32]. To determine between-person differences, the mean of each outcome over the two time points for each participant is regressed on the mean of each biomarker Z-score over two time points for each participant and resulted in cross-sectional estimates (between-persons association, formula 1). To determine within-person differences, the deviation of the Z-score at each time point (ie. baseline and follow-up) from the mean Z-score for each participant is regressed on the deviation of the outcome at each time point with the mean outcome for each participant. This results in estimates that show the association of change in biomarker Z-scores with change in outcome (within-persons association, formula 2).

$$\bar{y} \sim \bar{x} \tag{1}$$

$$y_{t=1,2} - \bar{y} \sim x_{t=1,2} - \bar{x} \tag{2}$$

We reported regression coefficients (betas) and 95% confidence intervals for the total

populations, and stratified for sex. Coefficients for the between-person associations are interpreted as the difference in echo outcomes between participants that differed with regard to biomarker Z-scores. For example, a coefficient of 1 for mean low-grade inflammation Z-score for LVEF would indicate that, on average, participants with 1 SD higher low-grade inflammation Z-score had LVEF that was 1 percentage points higher than participants with mean low-grade inflammation Z-score. Coefficients for the within-person associations are interpreted as the mean change in LVEF within a participant with a one SD increase in biomarker Z-score. For example, a coefficient of 1 for the deviation of low-grade inflammation Z-score for LVEF would indicate that a participants whose low-grade inflammation Z-score increased with 1 SD over time had an increase in LVEF of 1 percentage points.

Potential effect modification by sex was assessed using the 'double-demeaning' method [33]. In this method, the product of the determinant and the effect modifier is computed (e.g. low-grade inflammation Z-score $^*$ sex), and the individual-level mean of that product is subtracted. P-values for interaction <0.10 were considered significant.

**Confounders.** Potential confounders were selected a priori and included age, BMI smoking status, (residual) kidney function, hypertension status, CVD and anti-inflammatory medication use at baseline and follow-up. Anti-inflammatory medication can influence certain inflammatory markers and include nonsteroidal anti-inflammatory drugs (NSAIDs), platelet aggregation inhibitors (abciximab), lipid lowering agents (statins and niacins) and ACE inhibitors [34].

The first model (model 1) was adjusted for sex (not for the sex-specific models) and glucose metabolism status, and time-varying variables age and BMI. In the second model, time-varying variables estimated glomerular filtration rate (kidney function), hypertension status, history of CVD, current smoking status and medication use (NSAIDs, abciximab, statins, niacins and ACE inhibitors) were added. In the Hoorn Study, model 2 was additionally adjusted for time-varying HbA1c levels.

**Sensitivity analyses.** In sensitivity analyses, the Z-scores for the separate biomarkers were used as determinant in the sex-stratified mixed model analyses. In additional sensitivity analyses, Z-scores were computed with only biomarkers available in both cohorts (Hoorn and Flemengho): CRP, IL-6, IL-8, TNF-α and sICAM-1 for the low-grade inflammation Z-score and sICAM-1, sVCAM-1 and sE-selectin for the endothelial dysfunction Z-score to allow a direct comparison of the results over the cohorts. In a third sensitivity analysis, in the second model, we restricted the anti-inflammatory medication to NSAIDs only since NSAIDs lower inflammatory levels specifically.

## Results

The study population consisted of 383 and 491 participants for the Hoorn and FLEMENGHO Study, respectively (Table 1). Approximately half of the participants of both cohorts were female. Mean age of female participants was 66.9±5.4 and 49.4±14.7 years and male participants was 66.0±6.6 and 48.8±14.9 years, respectively. Prevalence of T2D was 29.2% and 3.5% and prior CVD was 47.5% and 4.1%, respectively. Mean follow-up time in the Hoorn study was 7.6±0.6 years and 4.8 [IQR: 4.4;5.2] years in FLEMENGHO.

Participants with follow-up measurements in the Hoorn Study were less often female, had a lower baseline prevalence of T2D and hypertension, had a lower baseline systolic blood pressure and had better baseline kidney function, cardiac structure (LVMI) and diastolic function (LAVI) than people without follow-up measurements (S1 Table). Participants with follow-up measurements in FLEMENGHO less often had a history of CVD and had worse baseline diastolic function (LAVI) than people without follow-up measurement (S1 Table).

**Table 1. Population characteristics, low-grade inflammation, endothelial dysfunction and echocardiographic measures for female and male participants of the Hoorn Study and FLEMENGHO.**

| | Hoorn Study (N = 383) | | | | FLEMENGHO (N = 491) | | | |
|---|---|---|---|---|---|---|---|---|
| | Female (N = 181) | Male (N = 202) | Baseline total | Follow-up total | Female (N = 248) | Male (N = 243) | Baseline total | Follow-up total |
| Age, years | 66.9±5.4 | 66.0±6.6 | 66.4±6.1 | 74.1±5.9 | 49.4±14.7 | 48.8±14.9 | 49.1±14.8 | 53.8±14.7 |
| BMI, kg/m$^2$ | 27.2±3.7 | 27.4±3.3 | 27.3±3.5 | 27.0±3.6 | 26.4±4.8 | 26.6±3.5 | 26.5±4.2 | 27.3±4.2 |
| GMS | | | | | | | | |
| NGM | 93 (51.4%) | 85 (42.1%) | 178 (46.5%) | 178 (46.5%) | N/A | N/A | N/A | N/A |
| IGM | 40 (22.1%) | 51 (25.3%) | 91 (23.8%) | 91 (23.8%) | N/A | N/A | N/A | N/A |
| T2D | 47 (26.0%) | 65 (32.2%) | 112 (29.2%) | 112 (29.2%) | 9 (3.6%) | 8 (3.3%) | 17 (3.5%) | 34 (6.9%) |
| SBP, mmHg | 139±21 | 139±18 | 138±19 | 144±18 | 126±17 | 129±13 | 127±16 | 131±16 |
| DBP, mmHg | 82±12 | 84±10 | 83±11 | 75±9.9 | 78±9 | 82±9 | 80±9 | 83±9 |
| Hypertension | 95 (52.5%) | 102 (50.5%) | 197 (51.4%) | 227 (59.3%) | 47 (19.0%) | 73 (30.0%) | 120 (24.4%) | 173 (35.2%) |
| Current smoker | 18 (9.9%) | 40 (19.8%) | 58 (15.1%) | 39 (10.2%) | 43 (17.3%) | 46 (18.9%) | 89 (18.1%) | 70 (14.3%) |
| eGFR, mL/min/1.73m$^2$ | 81.6±11.8 | 83.9±12.7 | 82.8±12.4 | 75±21 | 103±24.2 | 93.6±22.2 | 98±24 | 106±25 |
| History of CVD | 84 (46.4%) | 98 (48.5%) | 182 (47.5%) | 214 (55.9%) | 5 (2.0%) | 15 (6.2%) | 20 (4.1%) | 42 (8.6%) |
| Follow-up time, years | 7.6±0.6 | 7.7±0.6 | 7.6±0.6 | - | 4.7 [4.3;5.2] | 4.9 [4.4;5.2] | 4.8 [4.4;5.2] | - |
| *Low-grade inflammation* | | | | | | | | |
| CRP, mg/L | 2.0 [0.8;3.9] | 2.0 [1.1;3.9] | 2.0 [1.0;3.9] | 1.6 [0.8;3.7] | 1.5 [1.0;2.8] | 1.1 [0.9;1.6] | 1.2 [0.9;2.2] | 1.2 [0.9;2.1] |
| SAA, mg/L | 2.0 [1.3;3.1] | 1.3 [0.8;2.5] | 1.6 [1.0;2.9] | 1.7 [1.1;3.0] | N/A | N/A | N/A | N/A |
| IL-6, ng/L | 1.3 [1.0;2.2] | 1.4 [1.0;2.0] | 1.4 [1.0;2.1] | 1.6 [1.1;2.4] | 1.5 [1.0;2.4] | 1.3 [0.9;1.9] | 1.4 [1.0;2.0] | 1.4 [0.9;2.1] |
| IL-8, ng/L | 14.0 [11.5;19.1] | 13.2 [10.3;17.2] | 13.8 [10.8;18.2] | 10.3 [7.8;12.9] | 6.9 [5.2;9.7] | 7.0 [5.4;9.6] | 6.9 [5.3;9.6] | 9.8 [6.9;13.5] |
| sICAM-1, µg/L | 250±53 | 250±59 | 249±56 | 241±55 | 239±83 | 231±80 | 235±81 | 230 [189;287] |
| TNF-α, ng/L | 8.1 [7.0;9.6] | 8.1 [6.7;9.8] | 8.1 [6.8;9.7] | 8.3 [7.2;9.9] | 2.1 [1.8;2.6] | 2.2 [1.8;2.6] | 2.1 [1.8;2.6] | 2.5 [2.0;3.0] |
| *Endothelial dysfunction* | | | | | | | | |
| sICAM-1, µg/L | 250±53 | 250±59 | 249±56 | 241±55 | 239±83 | 231±80 | 235±81 | 230 [189;287] |
| sVCAM-1, µg/L | 374 [336;426] | 387 [342;438] | 381 [337;436] | 393±88 | 493±167 | 510±187 | 501±177 | 578±213 |
| sE-selectin, µg/L | 18.3±7.8 | 20.0±8.4 | 19.2±8.1 | 18.5±7.5 | 14.0 [10.2;17.9] | 16.0 [11.4;21.2] | 14.8 [10.8;19.4] | 15.3 [11.6;20.2] |
| sTM, µg/L | 3.4±0.7 | 3.6±0.9 | 3.5±0.8 | 3.8±1.0 | N/A | N/A | N/A | N/A |

Values are depicted as numbers (percentages); means±standard deviations; medians [interquartile ranges].

Abbreviations: FLEMENGHO = Flemish Study on Environment, Genes and Health Outcomes, BMI = body mass index, GMS = glucose metabolism status, NGM = normal glucose metabolism, IGM = impaired glucose metabolism, T2D = type 2 diabetes, SBP = systolic blood pressure, DBP = diastolic blood pressure, eGFR = estimated glomerular filtration rate, CVD = cardiovascular diseases, CRP = C-reactive protein, SAA = serum amyloid A, IL-6 = interleukin-6, IL-8 = interleukin-8, sICAM1 = soluble intercellular adhesion molecule 1, TNFa = tumor necrosis factor α, sVCAM1 = soluble vascular adhesion molecule 1, sE-selectin = soluble endothelial selectin, sTM = soluble thrombomodulin.

We did not observe consistent statistically significant within-person and between-persons associations of overall low-grade inflammatory and endothelial dysfunction biomarker Z-scores with echocardiographic measures of cardiac structure and function, neither in the Hoorn Study nor in FLEMENGHO (Table 2). Sex was an effect modifier in both cohorts for the association between endothelial dysfunction and LVEF (P-values for interaction 0.08 and 0.06 for model 2, respectively). For the within-person associations in the Hoorn Study, women whose endothelial dysfunction worsened with 1-SD over time had a decrease in LVEF of 3.7 (-7.4;0.1) percentage points. For the within-person association in men, participants whose endothelial dysfunction worsened with 1-SD over time had an increase in LVEF of 1.3 (-1.6;4.1) percentage points. For the within-person associations in FLEMENGHO, women

**Table 2. Within person and between person associations of low-grade inflammation or endothelial dysfunction on cardiac structure and function measures in the Hoorn Study and FLEMENGHO.**

| | Hoorn | | | | | | FLEMENGHO | | | | | |
|---|---|---|---|---|---|---|---|---|---|---|---|---|
| | Total (N = 289) | | Female (N = 137) | | Male (N = 152) | | Total (N = 315) | | Female (N = 159) | | Male (N = 156) | |
| **LVEF, %** | Within person | Between persons | Within person | Between persons | Within person | Between persons | Within person | Between persons | Within person | Between persons | Within person | Between persons |
| *Low-grade inflammation* | | | | | | | | | | | | |
| Model 1 | -0.4 (-2.4;1.7) | -0.7 (-2.3;0.9) | -1.7 (-4.9;1.4) | -0.8 (-3.1;1.6) | 0.8 (-2.0;3.5) | -0.6 (-2.8;1.7) | 0.3 (-1.0;1.6)* | -0.1 (-1.5;1.3)* | 1.2 (-0.6;3.0) | -0.2 (-2.1;1.6) | -0.4 (-2.2;1.4) | 0.1 (-2.0;2.2) |
| Model 2 | -0.5 (-2.6;1.7) | -0.4 (-2.0;1.2) | -1.2 (-4.5;2.2) | -0.5 (-2.8;1.7) | 1.0 (-1.7;3.7) | -0.1 (-2.3;2.0) | 0.02 (-1.3;1.3) | 0.04 (-1.4;1.4) | 0.5 (-1.5;2.4) | -0.3 (-2.1;1.5) | -0.6 (-2.4;1.2) | 0.4 (-1.8;2.5) |
| *Endothelial dysfunction* | | | | | | | | | | | | |
| Model 1 | -0.6 (-2.9;1.7)* | 1.2 (-0.2;2.6)* | -3.3 (-7.1;0.5) | 2.0 (-0.2;4.2) | 1.3 (-1.6;4.1) | 0.8 (-1.0;2.7) | -0.5 (-1.5;0.5)* | -0.4 (-1.3;0.5)* | 0.6 (-1.1;2.3) | -0.7 (-2.1;0.8) | -1.1 (-2.2;0.05) | -0.2 (-1.4;1.1) |
| Model 2 | -0.6 (-2.9;1.7)* | 1.0 (-0.4;2.4)* | -3.7 (-7.4;0.1) | 1.5 (-0.7;3.7) | 1.3 (-1.6;4.1) | 0.5 (-1.3;2.3) | -0.6 (-1.6;0.3)* | -0.4 (-1.4;0.5)* | 0.2 (-1.5;1.9) | -0.7 (-2.1;0.8) | **-1.2 (-2.3;-0.01)** | -0.2 (-1.5;1.1) |
| | Total (N = 304) | | Female (N = 146) | | Male (N = 158) | | Total (N = 477) | | Female (N = 242) | | Male (N = 235) | |
| **LVMI, g/m$^{2.7}$** | Within person | Between persons | Within person | Between persons | Within person | Between persons | Within person | Between persons | Within person | Between persons | Within person | Between persons |
| *Low-grade inflammation* | | | | | | | | | | | | |
| Model 1 | 0.2 (-2.0;2.3) | 0.3 (-1.5;2.2) | 0.1 (-2.9;3.1) | 1.4 (-1.2;3.9) | 0.3 (-2.8;3.5) | -0.7 (-3.4;2.1) | 0.5 (-0.5;1.5) | -0.4 (-1.9;1.0) | 0.4 (-0.9;1.7) | -0.3 (-2.1;1.6) | 0.6 (-0.9;2.0) | -0.6 (-2.9;1.6) |
| Model 2 | 0.2 (-2.1;2.4) | -0.6 (-2.4;1.2) | 0.2 (-2.8;3.3) | 0.6 (-1.8;2.9) | 0.3 (-3.0;3.5) | -1.9 (-4.5;0.8) | 0.5 (-0.5;1.4) | -0.6 (-2.0;0.9) | 0.4 (-0.9;1.7) | -0.2 (-2.0;1.6) | 0.5 (-0.9;1.9) | -0.5 (-2.7;1.8) |
| *Endothelial dysfunction* | | | | | | | | | | | | |
| Model 1 | 0.9 (-1.6;3.4) | -0.3 (-1.9;1.3) | -0.1 (-3.8;3.6) | 2.3 (-0.3;4.8) | 1.8 (-1.7;5.3) | -1.9 (-4.1;0.3) | 0.1 (-0.5;0.8) | -0.6 (-1.5;0.4) | 0.2 (-0.8;1.3) | -0.3 (-1.6;1.1) | 0.1 (-0.8;0.9) | -0.9 (-2.2;0.4) |
| Model 2 | 0.7 (-1.9;3.3) | -0.4 (-1.9;1.2) | -0.3 (-3.9;3.3) | 2.0 (-0.4;4.4) | 2.3 (-1.3;5.9) | -1.8 (-3.9;0.2) | 0.1 (-0.6;0.7) | -0.8 (-1.8;0.1) | 0.2 (-0.8;1.3) | -0.7 (-2.0;0.7) | 0.1 (-0.8;0.9) | -1.1 (-2.4;0.2) |
| | Total (N = 318) | | Female (N = 150) | | Male (N = 168) | | Total (N = 484) | | Female (N = 245) | | Male (N = 238) | |
| **LAVI, mL/m$^2$** | Within person | Between persons | Within person | Between persons | Within person | Between persons | Within person | Between persons | Within person | Between persons | Within person | Between persons |
| *Low-grade inflammation* | | | | | | | | | | | | |
| Model 1 | -0.6 (-2.3;1.2) | 1.0 (-0.9;3.0) | 0.7 (-1.9;3.2) | -0.8 (-3.3;1.8) | -1.8 (-4.3;0.6) | 2.2 (-0.6;5.0) | -0.1 (-0.8;0.6) | -0.6 (-1.6;0.5) | -0.2 (-1.0;0.7) | -0.7 (-2.0;0.6) | -0.2 (-1.3;0.9) | -0.3 (-2.0;1.4) |
| Model 2 | -0.6 (-2.5;1.2) | 0.8 (-1.2;2.7) | 1.1 (-1.6;3.9) | -0.9 (-3.6;1.7) | -2.3 (-4.7;0.1) | 1.9 (-1.0;4.8) | 0.01 (-0.7;0.7) | -0.5 (-1.5;0.6) | 0.02 (-0.8;0.9) | -0.6 (-1.9;0.6) | -0.1 (-1.2;0.9) | -0.3 (-2.0;1.4) |
| *Endothelial dysfunction* | | | | | | | | | | | | |
| Model 1 | 0.1 (-1.9;2.2) | 0.9 (-0.8;2.6) | 0.3 (-3.0;3.6) | 0.1 (-2.3;2.5) | -0.01 (-2.6;2.6) | 1.2 (-1.2;3.5) | 0.3 (-0.2;0.7) | -0.1 (-0.8;0.6) | 0.4 (-0.3;1.2) | -0.2 (-1.2;0.7) | 0.2 (-0.5;0.8) | -0.1 (-1.1;1.0) |
| Model 2 | -0.1 (-2.2;2.0) | 0.8 (-1.0;2.5) | 0.5 (-2.9;3.9) | 0.1 (-2.4;2.5) | -0.1 (-2.7;2.4) | 1.2 (-1.2;3.6) | 0.3 (-0.2;0.8) | -0.2 (-0.9;0.4) | 0.5 (-0.2;1.2) | -0.4 (-1.4;0.5) | 0.2 (-0.4;0.8) | -0.3 (-1.4;0.7) |

Results are expressed as unstandardized beta's with 95% confidence intervals. Model 1 is adjusted for sex (for the total populations), time-varying covariates age and BMI, and glucose metabolism status at baseline. Model 2 is additionally adjusted for time-varying covariates eGFR, hypertension, smoking status, medication use and CVD. In the Hoorn Study, Model 2 is additionally adjusted for time-varying HbA1c. Significant effect modification by sex (P <0.10) is denoted with *.

Abbreviations: LVEF = left ventricular ejection fraction, LVMI = left ventricular mass index, LAVI = left atrial volume index, BMI = body mass index, eGFR = estimated glomerular function, CVD = cardiovascular disease

whose endothelial dysfunction worsened with 1-SD over time had an increase in LVEF of 0.2 (-1.5;1.9) percentage points. For the within-person association in men, participants whose endothelial dysfunction worsened with 1-SD over time had a decrease in LVEF of 1.2 (-2.3;-0.01) percentage points.

In the Hoorn Study, effect modification by sex was apparent for TNF-α in relation to LVEF (P-for interaction = 0.01). For the within-person associations, women whose TNF-α levels increased with 1-SD over time had a decrease in LVEF of 2.2 (-4.5;0.01) percentage points. For the between-persons associations, on average, women with 1-SD higher TNF-α levels had a LVEF that was 0.6 (-1.9;0.7) percentage points lower than women with mean TNF-α levels. For the within-person associations in men, participants whose TNF-α levels increased with 1-SD over time had an increase in LVEF of 2.2 (-0.01;4.3) percentage points. For the between-persons associations, on average, men with 1-SD higher TNF-α levels had a LVEF that was 0.6 (-1.9;0.8) percentage points lower than men with mean TNF-α levels (S2 Table). Effect modification by sex was also apparent for SAA with LAVI (P-for interaction = 0.09). For the within-person associations, women whose SAA levels increased with 1-SD over time had an increase in LAVI of 0.3 (-1.5;2.1) mL/m$^2$. For the between-persons associations, on average, women with 1-SD higher SAA levels had a LAVI that was 0.4 (-1.5;2.2) mL/m$^2$ higher than women with mean SAA levels. For the within-person associations in men, participants whose SAA levels increased with 1-SD over time had a decrease in LAVI of 1.6 (-2.7;-0.4) mL/m$^2$. For the between-persons associations, on average, men with 1-SD higher SAA levels had a LAVI that was 0.2 (-1.5;1.8) mL/m$^2$ higher than men with mean SAA levels (S2 Table).

In FLEMENGHO, effect modification by sex was apparent for sICAM-1 and E-selectin in relation to LVEF (P-for interaction = 0.03 and 0.05, respectively). For the within-person associations, women whose sICAM-1 or E-selectin levels increased with 1-SD over time had an increase in LVEF of 0.4 (-1.4;2.1) and 0.1 (-1.9;2.0) percentage points, respectively. For the between-persons associations, on average, women with 1-SD higher sICAM-1 or E-selectin levels had a LVEF that was 0.5 (-1.9;0.9) and 0.3 (-1.4;0.9) percentage points lower than women with mean sICAM-1 or E-selectin levels. For the within-person associations in men, participants whose sICAM-1 or E-selectin levels increased with 1-SD over time had an decrease in LVEF of 1.2 (-2.2;-0.2) and 1.6 (-2.7;-0.5) percentage points, respectively. For the between-persons associations, on average, men with 1-SD higher sICAM-1 or E-selectin levels had a LVEF that was 0.2 (-1.4;1.1) and 0.1 (-1.2;1.0) percentage points lower than men with mean sICAM-1 or E-selectin levels (S2 Table).

Restricting the low-grade inflammation and endothelial dysfunction Z-scores to biomarkers available in both cohorts did not affect the associations (S3 Table). Adjustment for use of NSAIDs at baseline instead of use of all inflammatory medication did not materially change our results.

## Discussion

Our study did not show consistent associations of markers of low-grade inflammation or endothelial dysfunction with cardiac structure and function, nor any evidence of effect modification by sex. Nevertheless, some evidence of effect modification was present for specific biomarkers. An increase of TNF-α over time was associated with a decrease in LVEF at follow-up in female participants of the Hoorn study, but not in male participants. In addition, an increase of E-selectin over time was associated with a decreased LVEF at follow-up in the FLEMENGHO Study.

In the current study, the overall low-grade inflammation and endothelial dysfunction or separate biomarker Z-scores were longitudinally not associated with echocardiographic

measures of cardiac structure and function in the total populations. Contrary to our results, only one longitudinal study in hypertensive patients with chronic kidney disease showed that higher IL-6 at baseline was associated with increased LVMI after 3 years of follow-up [35]. However, in that study, repeated measurements of IL-6 and other inflammatory biomarkers were not available. Therefore that study does not reflect possible longitudinal changes of inflammatory biomarkers and their effect on cardiac structure and function.

Results on the cross-sectional associations between inflammatory markers and measures of cardiac structure and function are inconsistent. Other studies did not find any relationship between inflammatory markers with measures of cardiac structure and function. However, these studies were all performed in specific patient populations with either hypertrophic cardiomyopathy or stable HFpEF and did not adjust for confounding factors [36, 37]. Studies that did find associations between inflammatory markers and DD or HFpEF, either compared HFpEF with HFrEF, were case-control studies, or were performed in young African Americans [11, 12, 15]. A potential explanation for the findings in our study could be that we used two population-based cohorts instead of a more extreme comparison of HFpEF-patients with healthy controls. This is also illustrated by the levels of inflammatory markers, such as IL-6 and TNF-α, that were higher in populations with HFpEF than we observed in our study: values for IL-6 were 8.2 ng/L for patients with HFpEF in another study versus 1.4 ng/L for participants in our study, and for TNF-α 56.7 ng/L versus approximately 5.0 ng/L, respectively [11]. Consequently, as the participants in our cohorts are relatively healthy, this resulted in smaller association measures in comparison to studies that compare HFpEF patients with healthy controls.

Another explanation for our findings could be that biomarkers were measured in serum and not peripheral blood mononuclear cells or cardiac tissue [38]. The latter would reflect inflammation in the myocardium (tissue level) instead of systemic inflammation, resulted in measurement bias. Additionally, other mechanisms, such as fibrosis and apoptosis, could take place that lead to the development of DD and HFpEF. Moreover, not all low-grade inflammatory and endothelial dysfunction biomarkers contribute to this development [39, 40].

Overall, effect modification by sex was observed for the associations between endothelial dysfunction and LVEF, but stratified results over the two cohorts were inconsistent. A possible explanation is that participants in the Hoorn Study are older than the participants in FLEMENGHO (66 versus 49 years), which resulted in the inclusion of mainly menopausal women in the Hoorn Study. This could lead to a cardio protective effect of estrogens in younger women in FLEMENGHO [41]. For some individual biomarkers, we observed effect modification by sex, but this was not consistent, and resulted in inconsistent results in both cohorts.

Moreover, IL-6, which is hypothesized to be one of the inflammatory biomarkers that drives early cardiac changes [8], was not associated with echocardiographic measures in our cohorts. This is in line with another study in German women and men [20] with a similar age distribution as our cohorts. This alludes that in a general population other mechanisms also take place, such as fibrosis and apoptosis, which could lead to the development of DD and HFpEF. Alternatively, not all low-grade inflammatory and endothelial dysfunction biomarkers contribute to this development [39, 40].

This is the first study that determined the sex-specific longitudinal and between person association of low-grade inflammation and endothelial dysfunction with echocardiographic measures in a general population. The strengths of this study include the use of two general population cohorts and their longitudinal design with a follow-up duration between four and eight years. Furthermore, the use of a within-between model allowed us to distinguish between within-person and between-persons associations. This is not possible with a standard linear regression model, and allowed us to adjust for covariates that were measured at both baseline

and follow-up. Last, both cohorts have an extensive set of low-grade inflammation and endothelial dysfunction biomarkers measured at both baseline and follow-up in a standardized fashion.

There were also some limitations that needed to be discussed. First, results of both cohorts could not be pooled as biomarkers were measured differently and because of different inclusion criteria, such as oversampling of IGM and T2D in the Hoorn Study, and lower minimum age in FLEMENGHO Study. Second, our findings may be biased due to survival bias, as loss to follow-up occurred between the baseline and follow-up measurements in both cohorts: 43% and 24% in the Hoorn and FLEMENGHO Study, respectively. This could have resulted in a relatively healthy study sample, which, in combination with the relatively short follow-up times, may have hampered the detection of early changes in the HFpEF process. Finally, due to multiple comparisons some incidental associations within a certain cohort could be a chance finding as well.

## Conclusion

In conclusion, our study did not show consistent associations of markers of low-grade inflammation or endothelial dysfunction with echocardiographic measures of cardiac structure and function. We could not detect consistent effect modification by sex in these associations across both cohorts. As results from our study and other studies are inconsistent, future research should focus on biomarkers related to other mechanisms. For example, fibrosis and apoptosis that also play a role in the development of HFpEF in prospective cohorts.

## Supporting information

**S1 Table. Baseline characteristics of complete cases (N = 383/491) versus loss to follow-up (N = 346/156) of the Hoorn and FLEMENGHO Study participants.**
(PDF)

**S2 Table. 'Longitudinal' and between person associations of separate biomarkers on cardiac structure and function measures in the Hoorn Study and FLEMENGHO.**
(PDF)

**S3 Table. Sensitivity analyses of the 'longitudinal' and between person associations of low-grade inflammation or endothelial dysfunction on cardiac structure and function measures in the Hoorn Study and FLEMENGHO.**
(PDF)

## Author Contributions

**Conceptualization:** Sharon Remmelzwaal, A. Johanne van Ballegooijen.

**Data curation:** Sharon Remmelzwaal, Lutgarde Thijs.

**Formal analysis:** Sharon Remmelzwaal.

**Funding acquisition:** Joline W. J. Beulens, Stephane R. B. Heymans, A. Johanne van Ballegooijen.

**Investigation:** Sharon Remmelzwaal.

**Methodology:** Sharon Remmelzwaal.

**Project administration:** Joline W. J. Beulens, Stephane R. B. Heymans, A. Johanne van Ballegooijen.

**Supervision:** Joline W. J. Beulens.

**Visualization:** Sharon Remmelzwaal.

**Writing – original draft:** Sharon Remmelzwaal.

**Writing – review & editing:** Joline W. J. Beulens, Petra J. M. Elders, Coen D. A. Stehouwer, Zhenyu Zhang, M. Louis Handoko, Yolande Appelman, Vanessa van Empel, Stephane R. B. Heymans, Lutgarde Thijs, Jan A. Staessen, A. Johanne van Ballegooijen.

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
