## [Decision Letter · Decision Letter 0]

26 Mar 2021

PONE-D-21-05358

Sex differences in the longitudinal relationship of low-grade inflammation and echocardiographic measures in the Hoorn and FLEMENGHO Study

PLOS ONE

Dear Dr. Remmelzwaal,

Thank you for submitting your manuscript to PLOS ONE. After careful consideration, we feel that it has merit but does not fully meet PLOS ONE’s publication criteria as it currently stands. Therefore, we invite you to submit a revised version of the manuscript that addresses the points raised during the review process.

As you can see from the comments, both reviewers found the study of interest, but suggested some minor changes. I agree with the reviewers and feel following their suggestions would certainly improve the paper.

We look forward to receiving your revised manuscript.

Kind regards,

Harald Mischak

Academic Editor

PLOS ONE

Journal Requirements:

1. Please ensure that your manuscript meets PLOS ONE's style requirements, including those for file naming. The PLOS ONE style templates can be found athttps://journals.plos.org/plosone/s/file?id=wjVg/PLOSOne_formatting_sample_main_body.pdf andhttps://journals.plos.org/plosone/s/file?id=ba62/PLOSOne_formatting_sample_title_authors_affiliations.pdf

2. In ethics statement in the manuscript and in the online submission form, please provide additional information about the patient records/samples used in your retrospective study. Specifically, please ensure that you have discussed whether all data/samples were fully anonymized before you accessed them.

Additional Editor Comments (if provided):

Reviewers' comments:

Reviewer's Responses to Questions

**Comments to the Author**

1. Is the manuscript technically sound, and do the data support the conclusions?

Reviewer #1: Partly

Reviewer #2: Yes

2. Has the statistical analysis been performed appropriately and rigorously? 

Reviewer #1: Yes

Reviewer #2: Yes

3. Have the authors made all data underlying the findings in their manuscript fully available?

Reviewer #1: Yes

Reviewer #2: Yes

4. Is the manuscript presented in an intelligible fashion and written in standard English?

Reviewer #1: Yes

Reviewer #2: Yes

5. Review Comments to the Author

Reviewer #1: It is difficult to determine the exact echocardiographic methods employed. The references then reference another paper that references another paper, and I am not sure if the original reference is to support the use of the echocardiographic measurement or the methodology used. A few sentences would make this clearer to the reader. Suggestions are as follows using sentences copied from the echocardiographic methodology on page 8:

LVEF (%) was used as an index of left ventricular systolic function and was calculated by the modified Simpson’s rule(26) (Please specify single plane apical 4 chamber view or biplane apical 4 and 2 chamber views).

Left ventricular mass (Please state if this was done using M-Mode or 2D imaging, I think it was M-Mode, but I can't be sure) –indexed to height to the power of 2.7 (LVMI, g/m2.7 193 ) – was determined to assess cardiac structure.

LAVI (mL/m2) served as an index for LV diastolic function and was calculated by indexing single-plane maximum left atrial volume (please state apical 4 chamber view, if that was the view used) by body surface area.

The same methodology section states that mitral valve disease was excluded, which I found in the paper referenced, but I can't tell if aortic valve disease was also excluded. Please clarify.

Reviewer #2: I read article entitled “Sex differences in the longitudinal relationship of 2 low-grade inflammation and echocardiographic 3 measures in the Hoorn and FLEMENGHO Study”. I want to thank you for this good work. They did not show consistent associations of markers of low-grade inflammation or endothelial dysfunction with cardiac structure and function, nor any evidence of effect modification by sex. The authors could make diastolic function assessment in accordance with ASE / EACVI standards. However, it is a well written study. This paper is suitable for publication in its current form.

6. PLOS authors have the option to publish the peer review history of their article (what does this mean?). If published, this will include your full peer review and any attached files.

Reviewer #1: No

Reviewer #2: No

---

## [Author Response · Author response to Decision Letter 0]

19 Apr 2021

Ms. Ref. No.: PONE-D-21-05358

Title: Sex differences in the longitudinal relationship of low-grade inflammation and echocardiographic measures in the Hoorn and FLEMENGHO Study

Response to the editor and reviewers

We would like to thank the reviewers and editor very much for carefully reviewing our manuscript. Their review helped us to improve the manuscript. We have addressed all comments below, point-by-point. In addition, we have revised the manuscript and have highlighted the changes in yellow in the revised manuscript.

Editor: 

Reply: We renamed all supplemental files accordingly and updated the reference list according to the journal’s standards. 

2. In ethics statement in the manuscript and in the online submission form, please provide additional information about the patient records/samples used in your retrospective study. Specifically, please ensure that you have discussed whether all data/samples were fully anonymized before you accessed them.

Reply: We added the following information in the Methods (page 7, lines 160-161): ‘All data and samples from the study participants were fully anonymized before analyses.’

 

Reviewer #1: 

It is difficult to determine the exact echocardiographic methods employed. The references then reference another paper that references another paper, and I am not sure if the original reference is to support the use of the echocardiographic measurement or the methodology used. A few sentences would make this clearer to the reader. Suggestions are as follows using sentences copied from the echocardiographic methodology on page 8:

LVEF (%) was used as an index of left ventricular systolic function and was calculated by the modified Simpson’s rule(26) (Please specify single plane apical 4 chamber view or biplane apical 4 and 2 chamber views).

Left ventricular mass (Please state if this was done using M-Mode or 2D imaging, I think it was M-Mode, but I can't be sure) –indexed to height to the power of 2.7 (LVMI, g/m2.7) – was determined to assess cardiac structure.

LAVI (mL/m2) served as an index for LV diastolic function and was calculated by indexing single-plane maximum left atrial volume (please state apical 4 chamber view, if that was the view used) by body surface area.

The same methodology section states that mitral valve disease was excluded, which I found in the paper referenced, but I can't tell if aortic valve disease was also excluded. Please clarify.

Reply: We added the following details in the Methods (page 8, lines 191-199, and page 9, lines 200-201): ‘LVEF (%) was used as an index of left ventricular systolic function and was calculated from the apical four chamber view in the Hoorn study[24, 25] and apical four- and two-chambers views in FLEMENGHO[27], using the modified Simpson’s rule[28]. Left ventricular mass – measured in M-Mode and indexed to height to the power of 2.7 (LVMI, g/m2.7) – was determined to assess cardiac structure. LAVI (mL/m2) served as an index for LV diastolic function and in the Hoorn study was calculated by indexing single-plane maximum left atrial volume from the apical four chamber view by body surface area. In FLEMENGHO, left atrial dimensions were measured in 3 orthogonal planes (parasternal long, lateral, and supero-inferior axes) and LAVI was calculated using the prolate-elipsoid method[29] and was indexed to body surface area.’

In the current study we did not exclude on additional diseases, such as mitral valve disease, as those patients with valvular diseases were already excluded because of missing data in either inflammatory or echocardiographic measurements. 

References

24. van den Hurk K, Alssema M, Kamp O, Henry RM, Stehouwer CD, Diamant M, et al. Slightly elevated B-type natriuretic peptide levels in a non-heart failure range indicate a worse left ventricular diastolic function in individuals with, as compared with individuals without, type 2 diabetes: the Hoorn Study. European Journal of Heart Failure. 2010;12(9):958-65. doi: 10.1093/eurjhf/hfq119.

25. Henry RM, Paulus WJ, Kamp O, Kostense PJ, Spijkerman AM, Dekker JM, et al. Deteriorating glucose tolerance status is associated with left ventricular dysfunction--the Hoorn Study. The Netherlands journal of medicine. 2008;66(3):110-7. Epub 2008/03/20. PubMed PMID: 18349466.

27. Kloch-Badelek M, Kuznetsova T, Sakiewicz W, Tikhonoff V, Ryabikov A, Gonzalez A, et al. Prevalence of left ventricular diastolic dysfunction in European populations based on cross-validated diagnostic thresholds. Cardiovasc Ultrasound. 2012;10:10. doi: 10.1186/1476-7120-10-10. PubMed PMID: 22429658; PubMed Central PMCID: PMCPMC3351014.

28. Folland ED, Parisi AF, Moynihan PF, Jones DR, Feldman CL, Tow DE. Assessment of left ventricular ejection fraction and volumes by real-time, two-dimensional echocardiography. A comparison of cineangiographic and radionuclide techniques. Circulation. 1979;60(4):760-6. doi: 10.1161/01.CIR.60.4.760.

29. Lang RM, Badano LP, Mor-Avi V, Afilalo J, Armstrong A, Ernande L, et al. Recommendations for Cardiac Chamber Quantification by Echocardiography in Adults: An Update from the American Society of Echocardiography and the European Association of Cardiovascular Imaging. European Heart Journal - Cardiovascular Imaging. 2015;16(3):233-71. doi: 10.1093/ehjci/jev014. 

Reviewer #2:

I read article entitled “Sex differences in the longitudinal relationship of 2 low-grade inflammation and echocardiographic 3 measures in the Hoorn and FLEMENGHO Study”. I want to thank you for this good work. They did not show consistent associations of markers of low-grade inflammation or endothelial dysfunction with cardiac structure and function, nor any evidence of effect modification by sex. The authors could make diastolic function assessment in accordance with ASE / EACVI standards. However, it is a well written study. This paper is suitable for publication in its current form.

Reply: We thank the reviewer for their feedback. As we did not measure e’ at baseline and TR velocity at both baseline and follow-up, assessment of diastolic function in accordance with ASE / EACVI standard could not be possible in our two population-based cohorts.

---

## [Editor Report · Decision Letter 1]

21 Apr 2021

Sex differences in the longitudinal relationship of low-grade inflammation and echocardiographic measures in the Hoorn and FLEMENGHO Study

PONE-D-21-05358R1

Dear Dr. Remmelzwaal,

We’re pleased to inform you that your manuscript has been judged scientifically suitable for publication and will be formally accepted for publication once it meets all outstanding technical requirements.

Kind regards,

Harald Mischak

Academic Editor

PLOS ONE

Additional Editor Comments (optional):

The comment from the reviewer was completely addressed, there are no open issues remaining.
---

## [Editor Report · Acceptance letter]

26 Apr 2021

PONE-D-21-05358R1 

Sex differences in the longitudinal relationship of low-grade inflammation and echocardiographic measures in the Hoorn and FLEMENGHO Study 

Dear Dr. Remmelzwaal:

I'm pleased to inform you that your manuscript has been deemed suitable for publication in PLOS ONE. Congratulations! Your manuscript is now with our production department. 

Kind regards, 

on behalf of

Prof. Harald Mischak 

Academic Editor

PLOS ONE